# Weak Values and Two-State Vector Formalism in Elementary Scattering and Reflectivity—A New Effect[†]

## C. Aris Chatzidimitriou-Dreismann

Institute of Chemistry, Sekr. C2, Faculty II, Technical University of Berlin, D-10623 Berlin, Germany; dreismann@chem.tu-berlin.de

† This paper is based on the talk at the 7th International Conference on New Frontiers in Physics (ICNFP 2018), Crete, Greece, 4–12 July 2018.

**Abstract:** The notions of Weak Value (WV) and Two-State Vector Formalism (TSVF), firstly introduced by Aharonov and collaborators, provide a quantum-theoretical formalism of extracting new information from a system in the limit of small disturbances to its state. Here, we explore two applications to the case of non-relativistic two-body scattering with one body weakly interacting with its environment. We present a physically compelling analysis of a new quantum effect: momentum transfer deficit and an accompanying enhanced energy transfer; or, equivalently, an apparent mass-deficit of the struck body. First, incoherent inelastic neutron scattering (INS) from protons of $H_2$ molecules in C-nanotubes is investigated. The data of the $H_2$ translational motion along the nanotube shows that the neutron apparently exchanges energy and momentum with a fictitious particle with mass of 0.64 atomic mass units (a.m.u.), which is in blatant contradiction with the expected value of 2 a.m.u. Second, the same theory is applied to neutron reflectivity—which is elastic and coherent—from the interface of (single crystal) Si with $H_2O$-$D_2O$ liquid mixtures. The data shows a striking enhanced reflectivity in a wide range of momentum transfers, which is tantamount to a momentum-transfer deficit with respect to conventional expectations. However, these effects find a natural interpretation within the WV-TSVF theoretical analysis under consideration. In summary, both scattering effects contradict conventional theoretical expectations, thus also supporting the novelty of the theoretical framework of WV and TVSF. Additionally, it should be pointed out that the two dynamical variables in the interaction Hamiltonian of the theoretical model belong to two different physical bodies.

**Keywords:** weak values; two-state vector formalism; quantum-correlation effects in condensed matter; incoherent neutron scattering; nanoscale confinement; hydrogen-carbon compounds; proton quantum mobility; solid–liquid surface phenomena; neutron reflectivity; X-ray diffraction

---

## 1. Introduction

The counter-intuitive Elitzur–Vaidman effect [1] concerning interaction-free measurements (popularly known as "bomb tester") revealed, among other facts, the ability to experimentally obtain information about an object's presence in some spatial region without ever "touching" it. This is a paradox in the frame of classical physics. Namely, in successful interaction-free "bomb detections", no physical quantity—like energy, momentum, angular momentum, spin, force, etc.—has been exchanged between the object and the probe particle (e.g., a photon). However, this information cannot be gained "at no charge"; the "costs" of this information are provided by the photon's wavefunction. Consequently, the experimental verification of this novel quantum effect demonstrates that the quantum-mechanical wavefunction is a real physical quantity—and not just an auxiliary construct for the calculation of expectation values of quantum observables, which have to be compared with

results of actual measurements as obtained with real instruments. (For a quite different opinion, see Discussion below.)

Furthermore, the fundamental time-inversion symmetry of the Schrödinger equation was shown to be a crucial feature in the novel theory of Weak Values (WV) and the Two-State Vector Formalism (TSVF) of Aharonov and collaborators; cf. [2–7] and references therein. Based on this theory, new experiments were suggested and several novel quantum effects were discovered; cf. e.g., [4] and references therein.

Very recently, Aharonov et al. [8] provided a remarkably simple and clear example demonstrating the predictive power of the theory, revealing an "anomalous" momentum exchange between photons (or particles) passing through a Mach–Zehnder interferometer (MZI) and colliding with one of its mirrors. Here, the measured photon's (particle's) final state, being post-selected in a specific output of the MZI, plays a crucial and succinct role, which is captured by the Aharonov–Bergmann–Lebowitz rule [9]. In essence, the revealed effect is as follows: Although the photons (particles) collide with the considered mirror only from the inside of the MZI, they do not push the mirror outwards, but rather they somehow succeed to pull it in [8]; see Section 2.

Inspired by this remarkable theoretical result, we recently started investigating a possible experimental application of this theory in the context of (non-relativistic) incoherent neutron scattering off atoms and molecules in condensed matter [10]. Related experimental results obtained by incoherent inelastic neutron scattering (IINS, o simply INS [11]) and deep-inelastic neutron scattering (DINS)—also called neutron Compton scattering (NCS)—were presented and theoretically analyzed [10].

In this paper, we focus on the applicability of the new WV-TSVF theory. Firstly (in Section Section 4), we investigate a counter-intuitive experimental result of neutron scattering from $H_2$ in carbon nanotubes and related materials, as obtained with conventional INS with the aid of modern 2-*dimensional* neutron spectrometers, e.g., the spectrometer ARCS [12]. In short, the findings—if analyzed with standard theory—correspond to a striking strong *mass deficit* of the scattering objects. The INS-observations have no known conventional interpretation. Secondly (in Section 5), we investigate an effect of surface physics, i.e., neutron reflectivity—which is elastic and coherent—from the interface of a single-crystal Si wafer with $H_2O$-$D_2O$ liquid mixtures. These measurements show a striking *increased neutron reflectivity* in a wide range of momentum transfers, which is tantamount to a momentum transfer deficit with respect to conventional expectations. We show that both effects appear to have natural interpretations within the WV-TSVF theoretical analysis under consideration.

Last but not least, a specific feature of the INS experiment under consideration should be pointed out here because (1) it affects several criticisms denying the novel character of WV and TSVF theory (not discussed in this paper), and (2) it is directly related with a particularly important point raised by Vaidman [13]. Concretely, in experiments done so far, WVs are measured using different degrees of freedom of the *same* quantum system. In our experiments, however, a WV referring to one body (i.e., the scattering H atom; or the nanoscopic layer at the Si/water interface) is observed via its interaction with a *different* quantum system (i.e., the meter, here: neutron). Hence, as Vaidman explicitly pointed out [13], in this case, the WV appears due to interference of a quantum entangled wave, thus having no analog in classical wave interference.

Weak values and the TSVF are nowadays active areas of research, providing new conceptual ideas and novel practical techniques for a wide range of applications. The present paper does not consider interpretational issues; instead, it proposes a new family of effects related with momentum-energy transfers, thus contributing to the connection between the "theory"-community and the community of experimental scattering physics.

## 2. Motivation

The central points of this paper concern (a) the measurements of momentum and energy transfers in real scattering experiments, (b) the associated predictions or expectations of conventional (classical or quantum) theory and, in particular, (c) the comparison of the experimental results with a new

theoretical prediction based on the theory of WV and TSVF. Point (c) has been intuitively motivated by certain intriguing results presented in a recent theoretical paper by Aharonov et al. [8].

Here, we consider a particular result of [8] only, which concerns "anomalous" momentum exchange between two quantum objects (i.e., a photon impinging on a small mirror of a MZI, combined with suitable post-selection); see Figure 1. This appears to have some analogy with the neutron-atom collision (or scattering) experiments considered below.

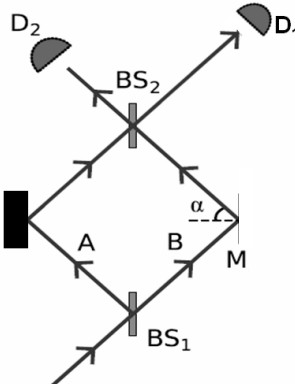

**Figure 1.** Mach–Zehnder interferometer and path of a light beam (or a particle, etc.). The mirror M on the right is a mesoscopic *quantum* object. Of particular interest are the photons (particles) emerging toward detector $D_2$ (adapted from Figure 2 of [8].)

Single photons (or particles) enter a device similar to a usual MZI, with the exception that one reflecting mirror is sufficiently small (say, a meso- or nanoscopic quantum object *M*) in order that its momentum distribution may be detectable by a suitable non-demolition measurement [14]. In [8], the authors derive the following astonishing result. Although the post-selected photons collide with the mirror *M* only from the inside of the MZI, they do not push *M* outwards, but rather *pull it in*. Obviously, this result does not have a classical interpretation.

Let us now consider a straightforward derivation of this effect, following closely the presentation of some derivations in [8] which are relevant for our purposes.

The two identical beamsplitters of the MZI have non-equal reflectivity $r$ and transmissivity $t$ (both real, with $r^2 + t^2 = 1$), and in the following we will assume $r > t$. Thus, when a single photon (particle) enters the MZI by impinging on the left side of the beamsplitter $BS_1$, the probabilities to be found in the arms $A$ and $B$ are $r^2$ and $t^2$, respectively.

The focus is on the momentum kick given to the mirror *M* due to each photon-*M* collision inside the interferometer, but only for photons emerging towards, or post-selected by, detector $D_2$. Using the standard convention, an incoming state $|\text{in}\rangle$ impinging on the beamsplitter $BS_1$ will emerge as a superposition of a reflected state $|R\rangle$ and a transmitted state $|T\rangle$; $|\text{in}\rangle \rightarrow ir|R\rangle + t|T\rangle$. Hence, when a single photon impinges from the left on $BS_1$, as illustrated in Figure 1, the effect of the beamsplitter is to produce inside the interferometer the state $|\Psi\rangle = ir|A\rangle + t|B\rangle$, where $|A\rangle$ and $|B\rangle$ denote the photon propagating along the arms $A$ and $B$ of the MZI, respectively.

The second beamsplitter, $BS_2$, is identical to the first. As one can readily check, a photon in the state $|\Phi_2\rangle = -ir|A\rangle + t|B\rangle$ emerges towards detector $D_2$. Furthermore, the probability of emerging towards detector $D_1$ is $|\langle\Phi_1|\Psi\rangle|^2 = 4r^2t^2$ while the probability of emerging towards $D_2$ is $|\langle\Phi_2|\Psi\rangle|^2 = (r^2 - t^2)^2 = 1 - 4r^2t^2$ [8].

Now, we send a *classical* light (particle) beam of intensity $I$ towards this interferometer. The light (particle) intensity in arm $A$ is then $I_A = r^2 I$ while the intensity in arm $B$ is $I_B = t^2 I$. The momentum given to the mirror $M$ by the beam inside the interferometer is $2I_B \cos\alpha = 2t^2 I \cos\alpha$. There is no doubt that this process pushes the mirror $M$ outward.

In contrast, let us now send a *quantum* beam of photons (particles) into the MZI. A photon when "going through arm A" will produce no kick to M. Each photon incident on $M$, which is in arm B, gives

$M$ a momentum kick $\delta$. (see [8] for important details concerning the momentum spread $\Delta$ of the mirror in relation to the kick $\delta$.)

Accordingly, if $\phi(p)$ is the initial quantum state of the mirror $M$ and $|\Psi\rangle$ is the quantum state of the photon after the input beamsplitter BS$_1$, but before reaching $M$, the reflection on $M$ results in the transition

$$|\Psi\rangle\phi(p) = (ir|A\rangle + t|B\rangle)\phi(p) \rightarrow ir|A\rangle\phi(p) + t|B\rangle\phi(p - \delta). \tag{1}$$

We shall now consider a single photon propagating through the interferometer. Given $\delta$ to be sufficiently small, we may approximate the total state of the photon and mirror just before the photon reaches the output beamsplitter BS$_2$ by

$$|\Psi\rangle\phi(p) \approx ir|A\rangle\phi(p) + t|B\rangle\left(\phi(p) - \frac{d\phi(p)}{dp}\delta\right) = |\Psi\rangle\phi(p) - t|B\rangle\frac{d\phi(p)}{dp}\delta. \tag{2}$$

Now, consider a photon emerging towards D$_2$, after passing beamsplitter BS$_2$. The state of the mirror $M$ is then given (up to normalization) by projecting the joint state onto the state of the photon corresponding to this beam, i.e.,

$$\langle\Phi_2|\left(|\Psi\rangle\phi(p) - t|B\rangle\frac{d\phi(p)}{dp}\delta\right). \tag{3}$$

A short derivation yields the result

$$\langle\Phi_2|\left(|\Psi\rangle\phi(p) - t|B\rangle\frac{d\phi(p)}{dp}\delta\right) = \langle\Phi_2|\Psi\rangle\phi(p - P_B^w\delta), \tag{4}$$

where $\mathbf{P}_B = |B\rangle\langle B|$ is the *projection operator* on state $|B\rangle$ and

$$P_B^w = \frac{\langle\Phi_2|\mathbf{P}_B|\Psi\rangle}{\langle\Phi_2|\Psi\rangle} \tag{5}$$

is the so-called *weak value* (WV) of $\mathbf{P}_B$ between the initial state $|\Psi\rangle$ and the final state $|\Phi_2\rangle$ [2–4,6]. Furthermore, the value of $P_B^w$ is readily found to be

$$P_B^w = \frac{\langle\Phi_2|\mathbf{P}_B|\Psi\rangle}{\langle\Phi_2|\Psi\rangle} = -\frac{t^2}{r^2 - t^2}; \tag{6}$$

see [8] for details of calculation. Hence, the momentum kick received by the mirror due to a photon emerging towards D$_2$ is

$$\delta p_M = P_B^w \delta = -\frac{t^2}{r^2 - t^2}\delta < 0. \tag{7}$$

The last inequality holds due to the case $r > t$ we are considering here.

The appearance of the WV of the projector $\mathbf{P}_B$ in Equation (4) a characteristic feature of the WV and TSVF theory. Namely, here the mirror represents a device measuring whether or not the photon is in arm $B$. The momentum of the mirror acts as a so-called pointer variable (no kick: the photon is in arm $A$; kick: the photon is in arm $B$). However, since the photon can only change the momentum of the mirror (pointer) by far less than its spread, we are in the so-called "weak measurement" regime, [2–6].

Moreover, in the particularly interesting case with $r > t$, the sign of the momentum received by the mirror is negative, hence the mirror is *pushed towards the inside* of the MZI. (Obviously, this also holds for the total momentum caused by all photons hitting $M$.) This astonishing momentum change is a result of the mirror receiving a quantum superposition between a kick $\delta$ and no kick at all, corresponding to the photon (particle) propagating through the two MZI-arms.

Obviously, the quantum-theoretical insight obtained here is dramatically different from the classical (or conventionally expected) one.

*Short Remarks on the General Theory*

Some additional remarks and/or explanations regarding the formal structure of WV-TSVF may be appropriate at this point. The starting point is a von Neumann-type impulsive interaction Hamiltonian $\hat{H} = c(t)\hat{A} \otimes \hat{M}$. For example, it may represent an impulsive collisional (or scattering) process. The real constant $g = \int c(t)dt$ is the strength of the interaction, which is assumed to have very short duration, so that the free evolution can be neglected; $\hat{A}$ is an observable of the system, and $\hat{M}$ is the meter variable of the measuring device (apparatus) that couples to the system. We proceed by following the presentation of Pati and Wu [15].

Before the interaction occurs, let the initial system-apparatus state be $\psi_i \otimes \Phi_i$. After the interaction, this state evolves as

$$\psi_i \otimes \Phi_i \rightarrow e^{-ig\hat{A} \otimes \hat{M}}\psi_i \otimes \Phi_i \tag{8}$$

(with $\hbar = 1$ for convenience). Now, the following point is a crucial element of TSVF: We post-select (with a strong measurement) a final state of the system, $\psi_f$. We are interested in the associated apparatus' final state, $\Phi_f$, from which the measuring result can be determined. $\Phi_f$ is obtained by tracing out the system's variables:

$$\Phi_f = \langle \psi_f | e^{-ig\hat{A} \otimes \hat{M}} | \psi_i \rangle \Phi_i \tag{9}$$

and, for a sufficiently weak interaction, one obtains (through linearization of the exponential)

$$\Phi_f \approx \langle \psi_f | 1 - ig\hat{A} \otimes \hat{M} | \psi_i \rangle \Phi_i = \langle \psi_f | \psi_i \rangle (1 - iA^w\hat{M})\Phi_i \approx \langle \psi_f | \psi_i \rangle e^{-igA^w\hat{M}}\Phi_i. \tag{10}$$

Here,

$$A^w = \frac{\langle \psi_f | \hat{A} | \psi_i \rangle}{\langle \psi_f | \psi_i \rangle} \tag{11}$$

is the WV of the system variable $\hat{A}$, which is a c-number; one assumes $\langle \psi_f | \psi_i \rangle \neq 0$. [Obviously, if $\psi_f = \psi_i$ then $A^w$ coincides with the expectation value of $\hat{A}$.] Thus the state of the measuring device evolves with an effective *one-body* Hamiltonian $\hat{H}_M = gA^w\hat{M}$, i.e., $\Phi_i \rightarrow e^{-igA^w\hat{M}}\Phi_i$.

To proceed, we make use of the formula (see e.g., [2], p. 39, or [15])

$$\hat{A}|\psi_i\rangle = \langle\hat{A}\rangle|\psi_i\rangle + \Delta\hat{A}|\overline{\psi}_i\rangle, \tag{12}$$

where $|\overline{\psi}_i\rangle$ is a state orthogonal to $|\psi_i\rangle$, $\langle\hat{A}\rangle = \langle\psi_i|\hat{A}|\psi_i\rangle$, and $\Delta\hat{A}$ is the uncertainty in the state $|\psi_i\rangle$, i.e., $(\Delta\hat{A})^2 = \langle\psi_i|(\hat{A} - \langle\hat{A}\rangle)^2|\psi_i\rangle$. Applying $\langle\psi_f|$ to both sides of Equation (12), we obtain

$$A^w = \langle\hat{A}\rangle + \Delta\hat{A}\frac{\langle\psi_f|\overline{\psi}_i\rangle}{\langle\psi_f|\psi_i\rangle} \equiv \langle\hat{A}\rangle + \delta A_w. \tag{13}$$

This result reveals a crucial reason for the difference $\delta A_w$ between the conventional expectation value $\langle\hat{A}\rangle$ and the WV: a non-vanishing uncertainty $\Delta\hat{A} > 0$ accompanied with a non-vanishing skalar product $\langle\psi_f|\overline{\psi}_i\rangle \neq 0$. In particular, the physical meaning of the latter is that the two states $|\psi_f\rangle$ and $|\overline{\psi}_i\rangle$ have the ability to produce quantum interferences.

In summary: these short derivations show that the novel features of WV and TSVF are caused by the interference between the post-selected state and another quantum state, which is orthogonal to the pre-selected state [15].

### 3. Elementary Neutron-Atom Scattering in View of WV and TSVF

*3.1. Model Hamiltonian*

In this theoretical section, we present the basic result of WV and TSVF applied to non-relativistic scattering processes, particularly in the physical context of *non-relativistic* neutron scattering, firstly proposed and explored in [10]. Here, we mainly follow the theoretical presentation in that paper.

The position and momentum of the probe particle (neutron) are denoted as $(q, p)$. Similarly, the position and momentum of the scattering system (atom, nucleus) are denoted as $(Q, P)$. $\hat{X}$ represents the operator of the corresponding observable $X$.

To simplify notations, in the following, it is sufficient to consider an effective *one-dimensional* problem along the direction of the collisional momentum transfer **K**, as given by conventional theory. The operator of atomic (nucleus) momentum component parallel to **K** is denoted with $\hat{P}$.

Due to momentum conservation in the two-body collision, one expects (*n*: neutron; *A*: atom)

$$- \hbar K_n = + \hbar K_A \equiv \hbar K > 0, \tag{14}$$

where $+\hbar K_A$ is the momentum transfer *on the atom* due to the collision. We choose $K_A$ to be positive, following standard notation (e.g., as in textbook [16] or review article [17]).

We proceed with the heuristic derivation of a von Neumann-type interaction Hamiltonian which captures momentum and energy transfers. Let the scattering atom be at rest before collision, that is $\langle \hat{P} \rangle_i = 0$. The elastic collision of a neutron with an atom with mass $M$ and initial momentum **P** results in the neutron's energy loss $E = E_i - E_f \equiv \hbar \omega_n > 0$. $E_i$ and $E_f$ are the neutron's initial and final kinetic energy, respectively. $E$ is transferred from the neutron to the struck atom which gains kinetic energy by

$$E = \frac{(\hbar \mathbf{K} + \mathbf{P})^2}{2M} - \frac{P^2}{2M} = \frac{(\hbar K)^2}{2M} + \frac{\hbar \mathbf{K} \cdot \mathbf{P}}{M}. \tag{15}$$

This is the so-called impulse approximation (IA) of standard theory [16,17]. Additionally, one may mention the relation $\hbar \mathbf{K} = \hbar \mathbf{k}_i - \hbar \mathbf{k}_f$, ($\mathbf{k}_i$ and $\mathbf{k}_f$ are the neutron's initial and final wavevectors, respectively), and the absolute value of momentum transfer, **K**, which in conventional theory [16,17] is:

$$|\mathbf{K}| = K = \sqrt{k_i^2 + k_f^2 - 2k_i k_f \cos \theta}. \tag{16}$$

The validity of Equation (15) is best demonstrated with a well known experimental INS-result: scattering from liquid He; see Figure 2 of [10] and references therein. The first term in the last right-hand-side of Equation (15), also called recoil parabola $(\hbar K)^2/(2M)$—M: mass of $^4$He atom—fits excellently the experimental scattering data points in the $K - E$ plane. The original measurements were carried out with the time-of-flight (TOF) spectrometer ARCS [12].

Having the theory of WV and TSVF in mind, one sees that the "larger" recoil term $(\hbar K)^2/(2M)$ may be viewed to result from a *strong* impulsive interaction (associated with momentum transfer $+\hbar K$ on the atom). In the IA holds $|K| \gg |P|$, and thus the "smaller" *Doppler* term $\hbar \mathbf{K} \cdot \mathbf{P}/M$ corresponds to a *weaker* interaction, in which the atomic momentum $\hat{P}$ couples with an appropriate dynamical variable of the neutron; the latter turns out to be $\hat{q}$ [10].

In view of the theory of WV and TSVF, the mentioned smaller Doppler term is expected to cause weak *deviations* from the conventionally expected momentum transfer $+\hbar K$. This can be formally captured by replacing $\hat{P}$ with a *small* momentum difference $\hat{P} - \hbar K \hat{I}_A$ ($\hat{I}_A$: identity operator in the atomic sub-space). For formal reasons, here is temporarily included a positive *smallness* factor, $0 < \lambda \ll 1$; see Discussion below. Summarizing, the proposed von Neumann-type model interaction Hamiltonian (describing the deviation from conventional theory) is [10]

$$\Delta \hat{H}_{int}(t) = +\lambda \, \delta(t) \, \hat{q} \otimes (\hat{P} - \hbar K \, \hat{I}_A). \tag{17}$$

($\delta(t)$: delta function). It should be emphasized that the *plus sign* in front of this expression is *not* arbitrary, since it is consistent with the aforementioned definitions (14) of momentum transfer; cf. [10].

The preceding physical motivation of the two parts of the model Hamiltonian of Equation (17) is in line with an associated example by Aharonov et al., which reads as follows: "Consider, for example, an ensemble of electrons hitting a nucleus in a particle collider. [...] The main interaction is purely electromagnetic, but there is also a relativistic and spin-orbit correction in higher orders which can be manifested now in the form of a weak interaction." ([18], p. 3.)

### 3.2. WV of Atomic Momentum Operator $\hat{P}$ and the Effect of "Anomalous" Momentum-Energy Transfer

In the following, the scattering atom represents the "system" of the general formalism, and the subscript $w$ in $X_w$ indicates that this is a WV of the observable $\hat{X}$. The WV of the identity operator is $(\hat{I}_A)_w = 1$, and thus for the WV of the atomic coupling operator $\hat{P} - \hbar K \hat{I}_A$ in the above interaction Hamiltonian:

$$(\hat{P} - \hbar K \hat{I}_A)_w = P_w - \hbar K. \tag{18}$$

To proceed, we first calculate the WV $P_w$ of $\hat{P}$ for some characteristic final state in momentum space. The derivation reveals a striking deviation—more precisely, a deficit—from the conventionally expected momentum transfer to the neutron. The latter represents the "pointer" of the general theory, and the pointer momentum variable $\hat{p}$ is conjugated to the neutron position $\hat{q}$.

For the calculation of the WV, it is natural to use the momentum space representation, because scattering experiments usually measure momenta (rather than the positions in real space) of the scatterers.

Let the atom initially be in a spatially confined state and at rest; e.g., in a potential representing physisorption on a surface or in a nanotube (as in the experiments of the next section). Usually, the initial atomic wavefunction $\Xi(P)_i$ is approximated by a Gaussian $G_A(P)$ centered at zero momentum,

$$\Xi(P)_i \approx G_A(P). \tag{19}$$

At sufficiently low temperatures, the atom will be in its ground state, and the width of $\Xi(P)_i$ is determined by the quantum uncertainty. The struck atom will move in the direction of momentum transfer $\hbar K_A = \hbar K$. Therefore, in the following calculations $P$ may represent the atomic momentum along the momentum transfer direction.

To facilitate the derivations and their physical meaning, let us make the following simplifying assumption concerning the final atomic state:

- The final atomic state has the same *width* in momentum space as the initial state.

This assumption is very common in molecular (optical) spectroscopy and/or quantum-chemical calculations of molecules. It captures the intuitive view that the impulsive transition is very fast (in the time scale of atomic or molecular motion) and so the atomic environment has not sufficient time to change configuration and to adapt to the "sudden disturbance" caused by the neutron–atom collision.

In other words, the final atomic momentum state should have the same shape as the initial state, but its center should be shifted from zero to the transferred momentum, i.e.,

$$\Xi(P)_f = \Xi(P - \hbar K_A)_i. \tag{20}$$

The WV of the atomic momentum operator $\hat{P}$ is calculated as follows:

$$\begin{aligned}
P_w &= \frac{\langle \Xi_f | \hat{P} | \Xi_i \rangle}{\langle \Xi_f | \Xi_i \rangle} = \frac{\int dP \, \Xi(P - \hbar K_A)_i \, P \Xi(P)_i}{\int dP \, \Xi(P - \hbar K_A)_i \, \Xi(P)_i}, \\
&= +\frac{\hbar K_A}{2} = +\frac{\hbar K}{2}.
\end{aligned} \tag{21}$$

The value of the integral in the numerator follows immediately from the following facts: (*a*) the two functions $\Xi$ are positioned symmetrically around the middle point $\bar{P} = \hbar K_A/2$ — one function is centered at 0, the other at $\hbar K_A$; and (*b*) in this integral $P$ is a linear factor. Note also that this result does not depend on the width of the Gaussian $\Xi$.

This is a quite surprising result because it is associated with a momentum-transfer correction of 50%; i.e., the scattered neutron measures a momentum kick being only half of the conventionally expected value, if one takes $\lambda = 1$; see also Discussion. In more detail, it holds that

$$(\hat{P} - \hbar K \, \hat{I}_A)_w = +\frac{\hbar K_A}{2} - \hbar K = -\frac{\hbar K}{2}. \tag{22}$$

To be more explicit, applying the well-known general result of WV and TSVF [2–4,6] we obtain for the *correction* to the shift of the meter's (i.e., the neutron's) pointer variable:

$$\langle \hat{p} \rangle_f - \langle \hat{p} \rangle_i = -\lambda \, (\hat{P} - \hbar K \, \hat{I}_A)_w = +\lambda \, \frac{\hbar K}{2}. \tag{23}$$

(See [10] for details.) Moreover, for the *total* momentum transfer shown by the pointer of the measuring device (here: neutrons), we have

$$\left[ \langle \hat{p} \rangle_f - \langle \hat{p} \rangle_i \right]_{\text{total}} = \left[ \langle \hat{p} \rangle_f - \langle \hat{p} \rangle_i \right]_{\text{conventional}} + \left[ \langle \hat{p} \rangle_f - \langle \hat{p} \rangle_i \right]_{\text{correction}}$$

$$= -\hbar K + \lambda \, \frac{\hbar K_A}{2}. \tag{24}$$

This expression represents the new quantum effect of *momentum-transfer deficit*: the absolute value of momentum transfer on the neutron predicted by the new theory is *smaller* than what is predicted by conventional theory:

$$| - \hbar K + \lambda \, \hbar K_A/2| < | - \hbar K|. \tag{25}$$

### 3.3. Plane Waves Approximation and Conventional Momentum Transfer

To illustrate the reason for this "anomalous" momentum transfer, let us make the widely used assumption of conventional neutron scattering theory (see e.g., [16,17]), which is:

- The final state of the struck atom should be a plane wave, i.e., it has vanishing width in momentum space—as especially assumed in the IA.

Interestingly, now the result of conventional theory follows straightforwardly within our formalism, i.e., the preceding correction due to WV and TSVF vanishes. Namely, the atomic final-state wavefunction is a delta function $\delta_A$ centered at the assumed transferred momentum $\hbar K_A$,

$$\Xi(P)_f = \delta_A(P - \hbar K_A). \tag{26}$$

The WV of $\hat{P}$ follows:

$$P_w = \frac{\langle \Xi_f | \hat{P} | \Xi_i \rangle}{\langle \Xi_f | \Xi_i \rangle} = \frac{\int dP \, \delta_A(P - \hbar K_A) \, P \, \Xi(P)_i}{\int dP \, \delta_A(P - \hbar K_A) \, \Xi(P)_i} = \frac{\hbar K_A \, \Xi(\hbar K_A)_i}{\Xi(\hbar K_A)_i}$$

$$= +\hbar K_A \equiv +\hbar K. \tag{27}$$

(Recall the notations of Equation (14).) Hence, the WV of the system coupling operator $(\hat{P} - \hbar K \, \hat{I}_A)$ becomes equal to zero,

$$(\hat{P} - \hbar K \, \hat{I}_A)_w = P_w - \hbar K = 0. \tag{28}$$

(Note that, according to standard quantum scattering theory, the scattered wave may acquire an additional phase factor, say $e^{i\chi}$, which does not affect the preceding result because this factor cancels out in the fractions of Equations (27) and (21).)

This result means that, under the usual assumption of "plane waves", the new theory yields *no correction* to the conventionally expected value of momentum transfer. Thus,

$$\left[\langle\hat{p}\rangle_f - \langle\hat{p}\rangle_i\right]_{\text{total}} = \left[\langle\hat{p}\rangle_f - \langle\hat{p}\rangle_i\right]_{\text{conventional}} = -\hbar K. \tag{29}$$

That is, the pointer of the measuring device—i.e., the experimentally determined momentum loss of a neutron—will show the conventionally expected value $-\hbar K$. Thus, the result of Equation (28) is consistent with conventional theory of the IA, and also of general incoherent neutron scattering theory. In other terms, the standard expectation of conventional theory is reproduced.

Comparison of the aforementioned two derivations shows that the magnitude of the momentum transfer "anomaly" under consideration depends on the "deformation" of the shape of the final atomic state. For example, if the final state is, say, "almost" a delta function (plane wave), then the anomalous momentum transfer deficit will be "very small"—and perhaps remain undetectable. Recall that the plane-wave assumption is standard in NCS and/or DINS data analysis; cf. [17].

## 4. Experimental Context—Incoherent Scattering

Note that instrumental details play a crucial role in possible applications of the theoretical framework under consideration because they concern pre-selection and, in particular, post-selection which is essential for TSVF.

### 4.1. Inelastic Neutron Scattering from Protons

Modern neutron scattering spectrometers are time-of-flight (TOF) instruments; cf. Figure 2. Here, a short pulse of neutrons produced at a neutron spallation source (e.g., SNS, Oak Ridge Nat. Lab., Tennessee, TN, USA) reaches the first monitor of the spectrometer, which triggers the measurement of TOF. Subsequently, a neutron scatters from the sample and may reach the detector, which stops the TOF measurement. In other terms, the experimentally determined values of momentum and energy transfers (which are the variables of the dynamic structure factor $S(K, E)$ of conventional theory [16,17]) are *not directly measured* (as one might believe) but calculated from the TOF-data by applying a strict momentum and energy conservation in each neutron–atom (or nucleus) binary collision [16,17].

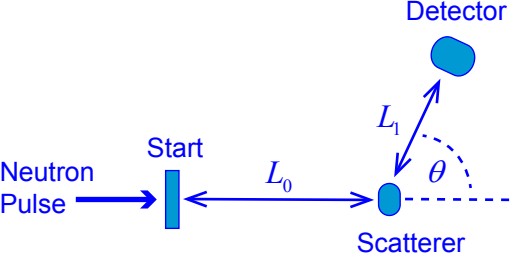

**Figure 2.** Schematic presentation of a time-of-flight neutron scattering setup. A detector, being a spatially localized system, performs a post-selection on the neutron's final state. (Taken from [10], with permission from *Quanta*.)

Modern spectrometers, like ARCS [12] (at SNS, Oak Ridge Nat. Lab., USA) have several thousand individual detectors (also called detector pixels). From each TOF-value measured with each individual detector at scattering angle $\theta$, the associated transfers of momentum ($\hbar K$) and energy ($E = \hbar\omega$) of the neutron to the struck particle are uniquely determined; see, e.g., [10]. Hence, a specific detector measures only one specific trajectory in the whole $K$–$E$ plane.

For a self-contained concise presentation of the TOF experimental procedure, see [10].

In this subsection, the preceding theoretical considerations and results are applied to concrete neutron scattering experiments, especially to *incoherent* scattering. In simple terms, "incoherent" means

that the impinging neutron (more generally: photon, electron, atom, etc.) collides with, and scatters from, a *single* particle (nucleus, atom, molecule, etc.). In the case of neutron scattering from protons (also referred to as H-atoms), the scattering is mainly incoherent due to the spin–flip mechanism of neutron–proton collision; see e.g., [16].

A clear first-principles explanation of *coherent* versus *incoherent* scattering may be found in the well-known Feynman Lectures [19], Section 3.3.

### 4.2. Effective Mass as Measured in the Scattering Experiment

In real experiments, deviations from the impulse approximation (IA) are well known; see, e.g., [10,17]. Within conventional theory, such deviations are understood as follows.

The energy conservation relation (15) for a two-body collision holds in the IA, which holds exactly for infinite momentum transfer and, consequently, for quasi-free scattering particles. However, it is not completely fulfilled at finite momentum and energy transfers of actual experiments, and thus so-called *final-state effects* (FSE) may become apparent [10,17]. (This term commonly includes both initial and final state effects). FSE are caused by environmental interactions with the struck particle, which affect both initial and final states of it. Here, we shall shortly describe this effect in the frame of conventional theory.

Firstly, as Feynman puts it: "We use the term *mass* as a quantitative measure of *inertia* ..." ([20], section 9-1). Therefore, when a scattering particle with mass $M$ is not completely free but *partially bound* to other adjacent particles, the impinging neutron seems to scatter from a particle with *higher* measure of inertia than $M$. This is because the adjacent massive particles are exerting forces on the scattering particle, which however can only hinder the motion of the scattering particle. In other terms, the particle is *dressed* by certain environmental degrees of freedom, and this dressing increases its inertia, i.e., its *effective* mass $M_{eff}$

$$M_{eff} \geq M \equiv M_{free}. \tag{30}$$

This reasoning corresponds to a well understood effect; cf. [10,17].

This effect can also be understood by referring to the aforementioned energy conservation relation, here including an additional term $E_{int} > 0$ describing the atom-environment interaction:

$$\bar{E} = \frac{\hbar^2 \bar{K}^2}{2M} + E_{int}, \tag{31}$$

where $\bar{E}$ and $\bar{K}$ refer to the center of a peak (measured by a specific detector). Thus, there will be a reduced amount of energy, $\bar{E} - E_{int}$, available as kinetic energy to the recoiling particle. It should be pointed out that $\bar{E}$ and $\bar{K}$ are determined from the kinematics of the neutron, in contrast to $E_{int}$, which is a quantity of the struck particle. If one tries to fulfill this equation with a pair $(E_{IA}, K_{IA})$ being determined by the conventional theory in the IA (ignoring FSE), one gets

$$E_{IA} = \frac{\hbar^2 K_{IA}^2}{2M_{eff}}. \tag{32}$$

Obviously, from the last two equations follows $M_{eff} > M$.

This effect can also be demonstrated with the aid of an exact calculation [17]. Assuming $K$-transfer to be fixed in a specifically designed experiment—a so-called *constant-K* measurement—the recoil (i.e., kinetic) energy of an atom with mass $M_{\text{eff}}$ will be *smaller* than predicted by the impulse approximation (in which the atom is free and has mass $M$). Figure 2 of Ref. [17] demonstrates this effect with the aid of an exact calculation of scattering from a harmonic oscillator.

An experimental demonstration of this effect is given by a DINS result of Sokol et al. [21], obtained from H atoms produced by chemisorbed, dissociated $H_2$ in the graphite intercalation compound (GIC) $C_8K$; see Figure 3. The experiments were done with an inverse geometry TOF spectrometer (of Argone Nat. Lab., USA) with energy selection in the *final* flight path. An accompanying DINS experiment

from the ionic solid KH was done with the same instrumental setup. Both measurements exhibited a remarkable similarity between these two H-recoil peaks [21]. Moreover, the measured E-transfer positions of the H peaks correspond to a very high effective mass, i.e., $M_{eff}(H) = 1.2$ a.m.u.

For more references about FSE and associated discussions of "effective mass" the interested reader may consult [10,17,22] and associated citations therein.

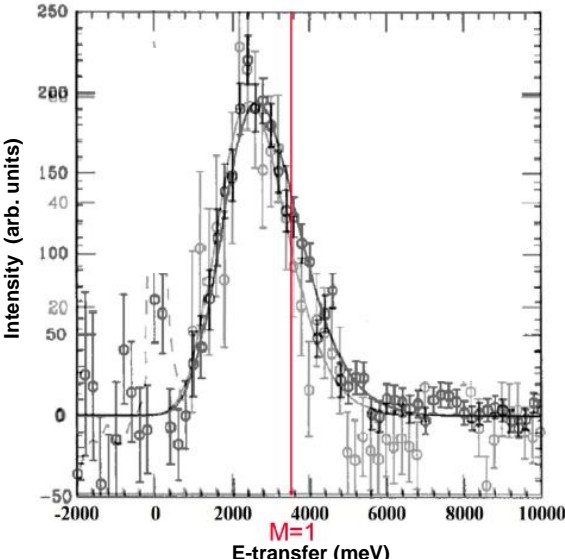

**Figure 3.** Deep-inelastic neutron scattering (DINS) from chemisorbed H in $C_8K$; data adapted from [21]. The maxima of the H peaks are about 2620 meV The vertical (red) line shows the position of the $M = 1$ a.m.u. recoil peak (at 3540 meV), assuming that the conventional theory, in particular the IA, is valid. Black points and fitted line: KH, Grey points and fitted line: $C_8K_{0.9}H_{0.13}$. The spectra were recorded with a selected final neutron energy $E_f = 4280$ meV. The peak maxima correspond to $M_{eff} = 1.2$ a.m.u. (Reproduced from Ref. [22], with permission from Institute of Physics (IOP).)

### 4.3. An Experiment on the New Scattering Effect—INS from Single $H_2$ Molecules in C-Nanotubes

In this section, the obtained theoretical results are compared with recent experiments. The derivations of the preceding section should apply to all neutron scattering subfields of interest (i.e., INS, NCS, DINS) as the derivations do not contain any specific assumption being valid in one subfield only. The presented experimental results may be considered as examples of the WV-TSVF-theoretical analysis of Section 3.

The revised physical intuitions offered by the theoretical result [8] outlined in Section 2 may be understood as the reason that led us to the analysis of the INS experiment considered here (and several others; see discussion below). In short: the measured scattering signal by a detector is due to a quantum superposition of the neutron (1) colliding with the atom (and thus giving a "positive" momentum transfer $\hbar K$ to the atom), and (2) being transmitted without collision, i.e., giving a "zero" momentum transfer—with the superposition of both causing the atom to receive an additional "negative" component $-\Delta(\hbar K)$ to its total momentum.

A remarkable INS result was observed by Olsen et al. [23] in the quantum excitation spectrum of $H_2$ adsorbed in multi-walled nanoporous carbon (with pore diameter about 8–20 Å).

The INS (or IINS) experiments were carried out at the new generation TOF spectrometer ARCS of Spallation Neutron Source SNS (Oak Ridge Nat. Lab., Tennessee, USA) [12]. In this experiment, the temperature was $T = 23$ K, and the incident neutron energy $E_i$ was 90 meV. The latter implies that the energy transfer cannot excite $H_2$-vibrations, but only rotation and translation (also called recoil) of $H_2$, which interacts only weakly with the substrate:

$$E = E_{rot} + E_{trans}. \tag{33}$$

The experimental two-dimensional incoherent inelastic neutron scattering intensity map $S(K, E)$ of H (after background subtraction) is shown in Figure 4, which is taken from the original paper [23]. The following features are clearly visible. First, the intensive peak centered at $E_{rot} \approx 14.7$ meV is due to the well known first rotational excitation $J = 0 \rightarrow 1$ of the $H_2$ molecule [11]. Furthermore, the wavevector transfer of this peak is $K_{rot} \approx 2.7$ Å$^{-1}$. Thus, the peak position in the $K$–$E$ plane shows that the experimentally determined mass of H that fulfills the relation $E_{rot} = (\hbar K_{rot})^2/2M_H$ is (within experimental error) the mass $M_H$ of the free H atom,

$$H_2 \text{ rotation } (J = 0 \rightarrow 1): \quad M_{eff} = M_H = 1.0079 \text{ a.m.u.} \tag{34}$$

(a.m.u.: atomic mass units.) In other words, the location of this rotational excitation in the $K$–$E$ plane agrees with conventional theoretical expectations for IINS, according to which each neutron scatters from a single H [11]. Recall that an agreement with conventional theory was also observed in the case of scattering from $^4$He; see above.

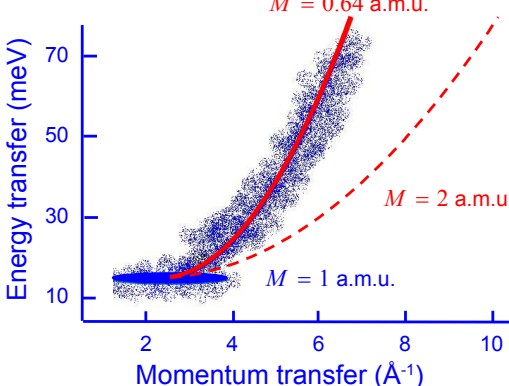

**Figure 4.** Experimental INS results from $H_2$ in carbon nanotubes, with incident neutron energy $E_i = 90$ meV; adapted from Figure 1 of Ref. [23]. The translation motion of the recoiling $H_2$ molecules causes the observed continuum of intensity, usually called "roto-recoil" (white-blue ribbon), starting at the well visible first rotational excitation of $H_2$ being centered at $E_{rot} \approx 14.7$ meV and $K_{rot} \approx 2.7$ Å$^{-1}$ (blue ellipsoid). The $K$–$E$ position of the latter is in agreement with conventional theory. In contrast, a detailed fit (red parabola; full line) to the roto-recoil data reveals a strong reduction of the effective mass of recoiling $H_2$, which appears to be only 0.64 a.m.u. (The red dashed line, right parabola) represents the conventional-theoretical parabola with effective mass 2 a.m.u.) For details of data analysis, see [23]. (Reproduced from Ref. [10], with permission from *Quanta*.)

Moreover, the authors provide a detailed analysis of the roto-recoil data from incoherent inelastic neutron scattering, as shown in Figure 4, and extracted from the data a strongly reduced effective mass of the whole recoiling $H_2$ molecule (left parabola):

$$H_2 \text{ translation (recoil): } \quad M_{eff}(H_2) \approx 0.64 \pm 0.07 \text{ a.m.u.} \tag{35}$$

This is in blatant contrast to the conventionally expected value $M(H_2) = 2.01$ a.m.u. for a freely recoiling $H_2$ molecule (right parabola).

An extensive numerical analysis of the data is presented in [23], being based on the analysis of the measured data within conventional theory [11,16].

This strong reduction of effective mass, which is far beyond any conceivable experimental error, corresponds to a strong reduction of momentum transfer by the factor 0.566. Namely, the observed momentum transfer deficit is about $-43\%$ of the conventionally expected momentum transfer. This provides first experimental evidence of the new *anomalous* effect of momentum-transfer deficit in an elementary neutron collision with a recoiling molecule.

Recall that, as explained above, every conventional $H_2$/substrate binding must increase the molecule's effective mass. Thus, these findings from IINS are in contrast to conventional (classical or quantum) theoretical expectations. However, they have a natural (albeit qualitative, at present) interpretation in the frame of WV and TSVF.

The above experimental results also show that the new two-dimensional spectroscopic technique, as provided by advanced TOF-spectrometers, e.g., ARCS, represents a powerful method that renders novel insights into quantum dynamics of molecules and condensed matter. Clearly, this is due to the fact that $K$ and $E$ transfers can be measured over a broad region of the $K$–$E$ plane. This advantage makes these new instruments superior to the older one-dimensional spectrometers, in which the detectors can only measure along a single specific trajectory in the $K$–$E$ plane, typically the recoil parabola of H.

## 5. Experimental Context—Reflectivity (Coherent Scattering)

Here, we present a new application of the theoretical result of Section 3 in the research field of neutron reflectometry, which belongs to *coherent elastic* neutron scattering.

The experimental method of *neutron reflectometry* (see e.g., [24]) is widely used for structural studies of various surfaces and interfaces on the nano-/mesoscopic length scale (say, thickness $d \approx 1$–$100$ nm). For example, investigations of physico-chemical and biological systems deal with air/liquid and solid/liquid interfaces, and new insights into the structure of such systems have been intensively studied. In such experiments, liquid water is often the dominant component of the materials analyzed, usually as the solvent.

Neutron reflectivity from Silicon/water interfaces, in particular from $Si/H_2O$-$D_2O$, at room temperature has been investigated in considerable detail [25]. The silicon wafer (single crystal) used had a hydrophobic coating, which protects the Si-interface and thus greatly facilitates the reproducibility of measurements. The experiments were carried out with the V6 neutron reflectometer at the BER II neutron source of the Berlin Neutron Scattering Center (BENSC), Hahn–Meitner Institute. The experimental reflectivity effect under consideration is shown in Figure 5.

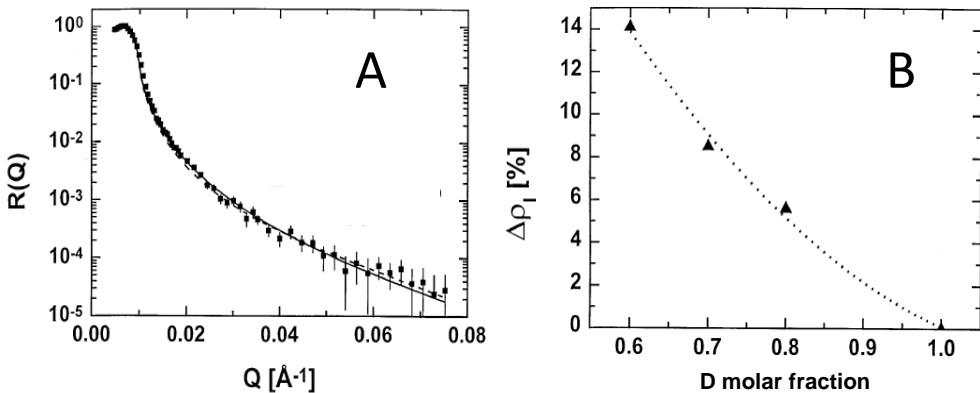

**Figure 5.** Experimental neutron reflectivity ($R(Q)$; log-scale) from Si/water interface. (**A**) measured and fitted reflectivity data for the Si/liquid interface of $H_2O$-$D_2O$ mixture with D molar fraction $x_D = 0.6$ at room temperature; (**B**) the observed "anomalous" increase of the scattering length density of the liquid at the solid–liquid interface for the additional layer models investigated in [25]: Relative deviation $\Delta\rho_I$ of the observed scattering length density $\rho_I$ of the (assumed) fictitious layer at the interface (with thickness $d_I \approx 50$ Å) from the conventionally expected scattering length density $\rho_{th}$ of the bulk liquid, for various $H_2O$-$D_2O$ mixtures. (Reproduced from [25], with permission from Elsevier.)

The analysis of the experimental results revealed the following surprising feature: The reflectivities $R(Q)$ ($\hbar Q$ being the momentum transfer) for the mixtures exhibit systematically increased values—in comparison to conventional theoretical expectations—for $Q$ higher than that of the total reflection edge. Described in conventional theoretical terms, this effect implies the presence of an *additional fictitious*

*layer* (between Si and bulk liquid) of thickness $d_I \approx 50$ Å with higher scattering length density than the bulk liquid, for all mixtures investigated.

This finding constitutes a new phenomenon, which among other aspects shows that one of the basic hypotheses in all previous H/D contrast variation works—i.e., the common assumption of homogeneity in the isotopic concentration in the liquid—is questionable [25].

In view of classical theory (cf. [24,25] for references), this effect is a signature of a hitherto unexplored physical chemistry of isotope mixtures near surfaces. However, in view of WV and TSVF, this effect finds a straightforward interpretation; it is a manifestation of the novel momentum-transfer deficit effect derived in Section 3. The nanoscale surface layer between Si-wafer and bulk water may be considered to correspond to the struck body (particle *A*) of the theoretical model. The physical momentum transfer by the specular reflection at the Si/water interface-layer is *not* the one calculated by the scattering angle $\theta$ (which is uniquely determined by the actual geometry of the source-sample-detector spatial configuration), but by the new WV-TSVF theory. The latter provides a correction to the classical-theoretical prediction, i.e., a reduction of the factual momentum transfer; see Section 3. As Figure 5A demonstrates, this *Q*-reduction corresponds to an *increased* reflectivity, which—in the frame of conventional theory—is caused by a (fictitious) increase of the scattering length density of the liquid at the interface; cf. Figure 5B.

## 6. A Preliminary XRD Result—Bragg Scattering

As stated in [8], WV and TSVF theory requires a complete revision of our intuition, which then can serve as a guide to finding novel quantum effects. Let us start with the following two points.

First, in view of the effects reported above, one may consider the reflectivity effect of the preceding section to represent an interference feature of possible neutron (Feynman) paths in the thin layered structure between bulk Si and bulk liquid.

Second, the quantum effect of Aharonov et al. [8] appears in a MZI; see Figure 1. Intuitively, and with some degree of abstraction, one may recognize here a "three-layer structure": the two mirrors define two layers and the two beamsplitters constitute the third layer. The photon (particle) "interacts" with this structure and is "scattered", and then is postselected in the direction determined by detector $D_2$.

These considerations lead to the speculative idea of possible similar "anomalies" that might be measurable in some well known multilayered structures: crystalline solids like e.g., silicon (Si).

The 100-year old XRD (X-ray diffraction) method, and the well known Bragg diffraction law, provide a means to test this intuitive idea. Very recently, we launched XRD investigations on some inorganic crystalline materials, e.g., on Si and $LaB_6$, both being cubic systems, which highly facilitates the analysis of the measured XRD data. ($LaB_6$ is the standard material used for calibration and other checks of any XRD spectrometer.) Figure 6 shows results of one experimental series during the last 12 months, using a transmission XRD spectrometer (system STADI P of STOE-GmbH, Darmstadt, Germany).

The preliminary findings show that the observed oscillating variation of the numerical value of the lattice parameter (the so-called lattice constant) of both materials is well reproducible; see Figure 6. These results are free of fitting parameters (which is only possible for cubic crystals). In view of the theory outlined in Section 3, this striking finding may be understood as follows: there is an unknown momentum-transfer variation in the "multi-layer scattering process" (i.e., in Bragg or Laue X-ray diffraction) that depends on the scattering angle $2\theta$ (and thus on the interlayer distance) of the observed diffraction peaks. Further experiments with different XRD instruments are under way.

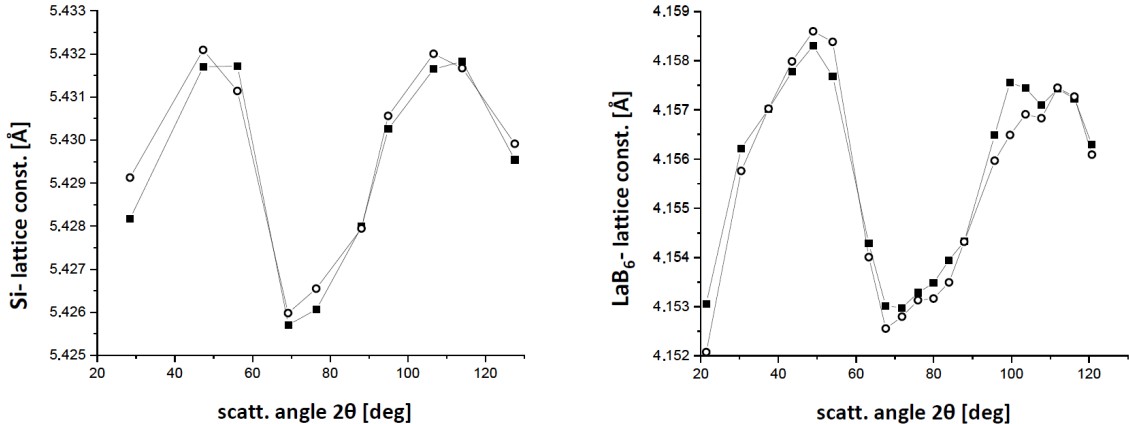

**Figure 6.** Preliminary results (left: Si, right: LaB$_6$)): Measured lattice constants of Si and LaB$_6$ (powders; both cubic), determined individually from the scattering angle 2$\theta$ of every Bragg peak appearing in the X-ray diffraction (XRD) data. That is, the results shown in the Figures contain no fitting parameter. Results of two independent experiments for each material are shown. The typical error of the determined values of the lattice constant is given by the difference between the two sets of results in each Figure.

## 7. Discussion

Here, we provide additional remarks which may further clarify the preceding theoretical and experimental results. Recall that only non-relativistic theory has been considered in the present paper.

Concerning the fundamental implication of the Elitzur–Vaidman effect stated in the Introduction, there exist differing opinions too. Here is an example: It is precisely because no physical quantity has been exchanged that provides the evidence that the wavefunction is not a physically real thing—it just conveys information, and is thus entirely a matter of inference. If one takes a Bayesian view of the wavefunction, for example, the interaction-free measurements can be viewed as simply learning whether the "bomb" is there or not, and the change of the wavefunction corresponds to simply changing one's mind about where the photon is, which is correlated with whether the "bomb is there", or not.

The *general* scheme of the derivations in Section 3 provides evidence that the new effect under consideration is of general character and thus may apply to any field of scattering physics, perhaps even to relativistic scattering in high-energy physics.

In the scientific literature, one sometimes meets the criticism that *post-selection* just means "throwing out some data". In the experimental context under consideration, however, post-selection means "performing a concrete measurement on the system, using a well defined detector system, and analyzing the measured data only". To explain this, consider the conventional expectation value of an observable $\hat{A}$, $\langle\psi|\hat{A}|\psi\rangle$, for a system being in the initial state $\psi$. Suppose $|\psi\rangle = \sum_j c_j|j\rangle$, where the kets $\{|j\rangle\}$ represent an orthogonal basis. The system and the measuring device are coupled weakly. Let us assume that we make a measurement of the observable $\hat{A}$ using the above basis and we make a post-section with respect to the final state $|j\rangle$. That is, we make a strong projective measurement corresponding to the projector $|j\rangle\langle j|$. If one is interested in the average (mean value) of the measuring results, one has:

$$\langle\psi|\hat{A}|\psi\rangle = \sum_j c_j^* \langle j|\hat{A}|\psi\rangle = \sum_j |c_j|^2 \frac{\langle j|\hat{A}|\psi\rangle}{\langle j|\psi\rangle} \equiv \sum_j |c_j|^2 A_{j,\psi}^w, \tag{36}$$

where $A_{j,\psi}^w$ is the WV of $\hat{A}$ with pre-selected state $|\psi\rangle$ and post-selected state $|j\rangle$, and $|c_j|^2$ is the probability for the occurrence of $|j\rangle$; cf. [6]. These relations show that (possibly existing) specific effects captured by some (measurable!) weak values $A_{j,\psi}^w$ may become "smeared out" in the average $\langle\psi|\hat{A}|\psi\rangle$. In simple terms: a "fine-grained" measurement may reveal more information than a "coarse-grained" one.

Of particular theoretical interest is the following point. The two operators $\hat{q}$ and $\hat{P}$ occurring in the von Neumann-type interaction Hamiltonian of Equation (17) refer to two *different* quantum systems. Consequently, as stressed by Vaidman [13], the concept of WV arises here due to the interference of a quantum *entangled* wave and therefore it has no analog in classical wave interference. This strongly supports the conclusion that WV is a genuinely *quantum* concept, and not some kind of "approximation".

In stark contrast, conventional non-relativistic neutron scattering theory treats the neutron as a *classical* mass point [16,17]; i.e., this theory contains only *c*-numbers referring to neutron's properties (e.g., the scattering length $b_A$).

The momentum-transfer deficit, and the associated effective-mass reduction of the scattering particle $H$ in the INS experiment, may appear as violating the energy and momentum conservation laws of basic physics. However, this is not the case because $H$ is *not* an isolated quantum system. It is helpful to write down the "conservation" relations

$$E_{H+env} = -E_n \quad \text{and} \quad \hbar K_{H+env} = -\hbar K_n, \tag{37}$$

which express energy and momentum conservation for the case that the *environment* (indicated with the subscript "*env*") of $H$ is not neglected. Hence, we may say that the quantum dynamics of the environment of $H$ is indispensable for the new WV-TSVF effect under consideration.

The INS results from $H_2$ in C-nanotubes [23] appear contradictory—in the frame of conventional theory—because: (i) the observed $J = 0 \rightarrow 1$ rotational excitation exhibits a $M_{eff} \approx 1$ a.m.u., see Equation (34), as conventionally expected. In contrast, (ii) the $M_{eff}$ of the observed translational motion of $H_2$ is not 2 a.m.u. as it should (because the whole $H_2$ undergoes translation), but only $M_{eff} \approx 0.64$ a.m.u. (see Equation (35)). This artificial "contradiction" just disappears in the light of the WV-TSVF theory because it implies (or, better: proves experimentally) that the protonic quantum *environments* are qualitatively *different* in cases (i) and (ii). In other terms: the WV-TSVF theory allows us to reveal and/or predict novel quantum effects of "system-environment" interactions that present-day quantum chemistry cannot describe.

Concerning the relevance of WV-TSVF (and WM) for the neutron scattering process, one may object that the neutron–nucleus potential (i.e., the conventional Fermi pseudo-potential [16]) is not "weak" in the specific sense of the new theory; see [10] for more discussion. Therefore, it is important to mention the recent generalization of the considered WV and WM theory by Oreshkov and Brun [26]. The authors showed that WM are universal, in the sense that any generalized measurement can be decomposed into a sequence of WMs. This important result is further supported by the work of Qin et al. [27], who showed that the main WM results can be extended to the realm of *arbitrary measurement strength*. Thus, the condition $\lambda \ll 1$ for the model Hamiltonian, Equation (17), may be dropped [10].

Very recently, some qualitatively new theoretical results of WV-TSVF have been obtained; e.g., the quantum Cheshire Cat effect [28], which has also been confirmed experimentally with neutron interferometry [29]. One may observe that these investigations also shed new light on the very notions of separability, quantum correlations and quantum entanglement [30].

In view of the experimental effect of Section 4, and in particular of the striking reduction of effective mass of H, it seems appropriate to mention here a speculative idea concerning the possible *practical* and/or *technological* relevance of the WV-TSVF theory: a more mobile (i.e., with smaller effective mass) H atom, or $H^+$ ion, enabled by a suitably chosen nanostructured fuel-cell material, would facilitate proton mobility and thus also improve the cell's efficiency.

## 8. Conclusions

The counter-intuitive momentum "exchange" taking place on a mirror of a MZI [8], as discussed in Section 2, and our results with neutron reflectometry in Section 5, suggest the idea that this effect could also be extended to the widely applied fields of experimental small-angle neutron scattering (SANS) and small-angle X-ray scattering (SAXS) on surfaces and/or thin multilayer structures of various materials. Additionally, the widely applied method of X-ray diffraction (XRD) might be a suitable tool for related investigations. Some preliminary XRD findings are shown in Section 6.

In conclusion, we believe that the novel theoretical formalism of WV and TSVF not only sheds new light on interpretational issues of fundamental quantum theory (like e.g., the time-inversion invariance of the basic physical laws, the meaning of the wavefunction, quantum entanglement, quantum discord, etc.), but it also offers a fascinating guide for our intuition to predict new effects, and even may help us to conceive and carry out related experiments, with the aim to reveal new quantum phenomena and to promote interdisciplinary research.

**Funding:** Part of this research was funded by European COST Action MP1403 (Nanoscale Quantum Optics).

**Acknowledgments:** I wish to thank Elisabeth Irran (Technical University of Berlin, TUB) for fruitful collaboration and assistance by the XRD measurements; and Philipp Stammer (TUB) and Ingmari Tietje (CERN) for helpful discussions.

**Conflicts of Interest:** The author declares no conflicts of interest.

## Abbreviations

The following abbreviations are used in this manuscript:

| | |
|---|---|
| WV | Weak Value |
| WM | Weak Measurement |
| TSVF | Two-State Vector Formalism |
| QE | Quantum Entanglement |
| MZI | Mach–Zehnder Interferometer |
| FSE | Final-State Effects |
| IA | Impulse Approximation |
| IINS | Inelastic Incoherent Neutron Scattering |
| INS | Equivalent to IINS |
| DINS | Deep-Inelastic Neutron Scattering |
| NCS | Neutron Compton Scattering, equivalent to DINS |
| TOF | Time-of-Flight |
| a.m.u. | Atomic Mass Unit |
| meV | Milli-Electron Volt |
| XRD | X-Ray Diffraction |

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
