# Peer review of "Weak Values and Two-State Vector Formalism in Elementary Scattering and Reflectivity—A New Effect"

_universe, doi:10.3390/universe5020058_

Reviewer 1 Report

Weak values and the two-state vector formalism are nowadays active areas of research, providing new conceptual ideas and novel practical techniques for a wide range of applications. In this context, the author is proposing a new family of effects related to momentum transfer deficit, which appears, for instance, in scattering processes. With this, the author does not only make a great service for this community, but also connects it with other communities. The presentation is very clear, the paper is well-written and the analysis is sound. I haven't found major flaws, but rather several interesting points and insights. I would therefore suggest to accept the paper as is. 

Author Response

I am really thankful to the Reviewer 1 for his/her supportive and encouraging remarks and the final recommendation!  

The Reviewer's comment  "... the author does not only make a great service for this community, but also connects it with other communities."  is a honour to me !

There is no specific scientific point to respond to.

The mentioned "minor spell check" has been done by three different persons.

Reviewer 2 Report

I enjoyed reading this manuscript. The style is clear and this is *very* appreciated. The only things that is missing, given that the weak value formalism is instead introduced very nicely, is an explanation of the two state vector formalism, which could come at the end of section 2. If the author included this, it would make for a very nice and complete paper.

I detected a typo on line 44 (possible)

Author Response

I am really thankful to the Reviewer 2 for his/her supportive and encouraging remarks and the final recommendation!   

Point 1:  There is one specific point to address. Reviewer 2 suggests  "....an explanation of the two state vector formalism, which could come at the end of section 2."

Response 1:

Responding positively, I wrote the new short Subsection 2.1 (on page 5, lines 132-150), being aware of the fact that "experimenters" by no means like to read papers with "long theoretical parts".

However, I hope that the non-specialized (in WV-TSVF) reader will appreciate this short (less than one page) theoretical  presentation and, more importantly, he/she may understand the novel physical feature & spirit of TSVF and WV, as expressed here. Namely, the formula (13) by Pati and Wu is short and contains onaly mathematical terms that experimenter s do already know. Of course, I am aware that this presentation is very limited and by no means covers all width and beauty of WV-TSVF.

However, if the Reviewer doesn't like it, for any reason, I can  simply cancel it.

The mentioned "minor spell check" has been done by three diffeerent persons.

Reviewer 3 Report

The author predicts some new effects in the momentum transfer in collisions between quantum particles.  The main effect concerns whether there is an additional "anomalous" momentum transfer.  The author shows that if the post collisions state of the atom is a plane wave (well defined momentum), then there is no anomalous momentum transfer.  On the other hand, if the wavefunction in momentum space is simply shifted by a constant amount, keeping its width fixed, the one obtains half of the momentum transfer.

In my view, the effect is not so surprising, since in the later case, there is a finite width in the pre and postselected state, so it is possible for a post-selected state to have momentum transfer asssociated with the uncertainty in the width of the wavefunction.

However, the author should also consider the possiblity of postselecting other final states - why does he only think about the one with a shifted average momentum?  In principle, any final state can be postselected, and it is only our bias of the outcoming state being one with a momentum shifted by K that selects this postselected state. 

It should remain true that if you average over all postselected states, that the average momentum transfer from the standard theory should be recovered.  Can the authors please present this result?

These points should be discussed more completely.

Also, I found the discussion in the introduction indicating that once must take the wavefunction to be real and objection to be without merit.  I fail to see how this conclusion is forced on us.

other typos:  line 163:  deep temperature; presumably you mean small temperature?

line 186, truck -> struck

line 274, the figure reference is broken.

In lines 410 - 413, I found these comments to be very far fetched.  After all, the pointer shift identified in this paper is multiplied by the small \lambda parameter, so this should be a very small effect.  Further, it only applied to postselected ensembles.

Author Response

-----------------------------------Point-by-point response---------------------------------------

Point 1:  In my view, the effect is not so surprising, since in the later case, there is a finite width in the pre and postselected state, so it is possible for a post-selected state to have momentum transfer associated with the uncertainty in the width of the wavefunction.

Response 1: I agree with all remarks concerning the *widths* of wavefunctions. However, I believe (as every "neutron scatterer" does) that the effect is surprising for the following plain point: In the experiments, the neutron seems to "see" only *one* side ot the system's momentum distribution.
-- that is, the side that leads to a *reduced" momentum transfer. Conventional theory (classical as well as quantum) would predict that the neutron must "see" the whole momentum distriution of the scatterer, thus exhibiting no "anomaly" at all.

Point 2:  However, the author should also consider the possiblity of postselecting other final states - why does he only think about the one with a shifted average momentum?  In principle, any final state can be postselected, and it is only our bias of the outcoming state being one with a momentum shifted by K that selects this postselected state. 

Response 2:  The remark " and it is only our bias of the outcoming state"  is not correct. Namely, in the INS experiment, all experimentally accesible  post-selected states --- associated with the wide range of measurable K-E plane by the spectrometer ARCS --- show this "anomaly", and therefore the *whole* roto-recoil parabola is shifted (and not only this centre or maximum etc).  Moreover, to both sides of the roto-recoil parabola is no signal, but onyl background --- see also the original paper 22.

Point 3:  It should remain true that if you average over all postselected states, that the average momentum transfer from the standard theory should be recovered.  Can the authors please present this result?

Response 3:  Following this request, I gladly added the requested result on page15, lines 387-xxx-392.(a ca 20 lines paragraph).  I thank the Reviewer 3 for this opportunity to make this point available to the non-specialised (in WV-TSVF) reader.

Moreover, the new Subsection 2.1 (requested by Reviewer 2) may provide additional help to the non-specialised reader who wish to see some  explicit and concrete detail causing the difference betwenn a measured WV and the well-known expectation value of some observable A.

[This remarkably short "explanation" is from Pati and Wu, in Ref 11] .

Point 4:  Also, I found the discussion in the introduction indicating that once must take the wavefunction to be real and objection to be without merit.  I fail to see how this conclusion is forced on us.

Response 4: To explain this point, I added a new text on page 1, lines 27-34. This explanation is fully in line with the famous proverb of R Landauer "Information is physical".  Here is the new  text I added:

    " Namely, in successful interaction-free "bomb detections",  no physical quantity --- like energy, momentum, angular momentum, spin, force, etc. ---  has been exchanged   between the object and the probe particle (e.g. a photon). However, this information cannot be  gained  "at no charge";  the "costs" of this information are provided by the photon's wavefunction. Consequently, the experimental verification of this novel quantum effect demonstrates   that the quantum-mechanical wavefunction is a real physical quantity --- and not just an      auxiliary construct for the calculation of expectation values of quantum observables, ...."

Points 5,6,7: typos, erroneous Figure citation:

Response to 5,6,7:  All corrected,    thank you for this help!

Point 8:  In lines 410 - 413, I found these comments to be very far fetched.  After all, the pointer shift identified in this paper is multiplied by the small \lambda parameter, so this should be a very small effect.  Further, it only applied to postselected ensembles.

Response 8:  The comment "very far  fetched"  may be right.  So I completely cancelled the corresponding lines 410-413 of the old version. That is, the revised manuscript (now on lines 425-430) doen't mention any batteries and/or Li-ions at all.  However, H-mobility is mentioned, since the INS experiment does revealed enhanced "mobility of H2 molecule". 

Additional Response: Moderate English changes have been done --- the majority of them are marked in a PDF of the old manuscritp I provide to the Assigned Editor.  Various typos, about 20 have been localized and corrected. Additionally, I would like to mention the associated remarks of Reviewers 1 and 2, who noticed:  "English language and style are fine ..."

Round  2

Reviewer 3 Report

After reviewing the revision, I am ok with the paper now being published.  However, I still have a problem with the author's statement:

" Namely, in successful interaction-free "bomb detections", no physical quantity --- like energy, momentum, angular momentum, spin, force, etc. --- has been exchanged between the object and the probe particle (e.g. a photon). However, this information cannot be gained "at no charge"; the "costs" of this information are provided by the photon's wavefunction. Consequently, the experimental verification of this novel quantum effect demonstrates that the quantum-mechanical wavefunction is a real physical quantity --- and not just an auxiliary construct for the calculation of expectation values of quantum observables, ...."

It is precisely because no physical change has been made that provides the evidence that the wavefunction is not a physically real thing - it just conveys information, and is thus entirely a matter of inference.  If one takes a Bayesian view of the wavefunction, for example, the IFM measurements can be viewed as simply learning whether the bomb is there or not, and the change of the wavefunction corresponds to simply changing one's mind about where the photon is, which is correlated with whether the bomb is there, or not.

Other than this minor change, it is fine that the paper is published.

Author Response

I thank very much the Reviewer 3 for his/her additional comments. Believing myself that the famous proverb of R. Landauer "Information is physical" has a deep meaning, I cannot change my own statement/viewpoint in the Introduction.     However, I included (almost verbatim) in Discussion the Reviewer's explicit statement/desciption in the report, which represents another scientific opinion (see Discussion, 2nd paragraph) .  Here is the new text in Discussion:

Concerning the fundamental implication of the Elitzur-Vaidman effect stated in the Introduction, there exist differing opinions too. Here is an example:             It is precisely because no physical quantity has been exchanged  that provides the evidence that the wavefunction is not a physically  real thing --- it just conveys information, and is thus entirely a matter of inference.    If one takes a Bayesian view of the wavefunction, for example, the interaction-free measurements  can be viewed as simply learning   whether the "bomb" is there or not, and the change of the wavefunction corresponds to simply changing one's mind about where the   photon is, which is correlated with whether the "bomb is there", or not.